# Toll-like Receptor 2 Mediates VEGF Overexpression and Mesothelial Hyperpermeability in Tuberculous Pleural Effusion

**DOI:** 10.3390/ijms24032846

**Published:** 2023-02-02

**Authors:** Wei-Lin Chen, Kai-Ling Lee, Kevin S. Lai, Jie-Heng Tsai, Shih-Hsin Hsiao, Chi-Li Chung

**Affiliations:** 1Department of Nursing, MacKay Junior College of Medicine, Nursing, and Management, Taipei 112, Taiwan; 2Graduate Institute of Clinical Medicine, College of Medicine, Taipei Medical University, Taipei 110, Taiwan; 3Division of Pulmonary Medicine, Department of Internal Medicine, Taipei Medical University Hospital, Taipei 110, Taiwan; 4School of Respiratory Therapy, College of Medicine, Taipei Medical University, Taipei 110, Taiwan; 5Division of Pulmonary Medicine, Department of Internal Medicine, School of Medicine, College of Medicine, Taipei Medical University, Taipei 110, Taiwan

**Keywords:** *Mycobacterium tuberculosis*, pleural effusion, pleural mesothelial cell, pleural mesothelial permeability, Toll-like receptor, tuberculous pleural effusion, vascular endothelial growth factor, zonula occludens

## Abstract

Toll-like receptor (TLR) is essential for the immune response to *Mycobacterium tuberculosis* (MTB) infection. However, the mechanism whereby TLR mediates the MTB-induced pleural mesothelial hyperpermeability in tuberculous pleural effusion (TBPE) remains unclear. Pleural effusion size and pleural fluid levels of vascular endothelial growth factor (VEGF) and soluble TLR2 (sTLR2) in patients with TBPE (n = 36) or transudative pleural effusion (TPE, n = 16) were measured. The effects of MTB H37Ra (MTBRa) on pleural mesothelial permeability and the expression of VEGF and zonula occludens (ZO)-1 in human pleural mesothelial cells (PMCs) were assessed. Levels of VEGF and sTLR2 were significantly elevated in TBPE compared to TPE. Moreover, effusion VEGF levels correlated positively, while sTLR2 values correlated negatively, with pleural effusion size in TBPE. In human PMCs, MTBRa substantially activated JNK/AP-1 signaling and upregulated VEGF expression, whereas knockdown of TLR2 remarkably inhibited MTBRa-induced JNK phosphorylation and VEGF overexpression. Additionally, both MTBRa and VEGF markedly reduced ZO-1 expression and induced pleural mesothelial permeability, while TLR2 silencing or pretreatment with anti-VEGF antibody significantly attenuated the MTBRa-triggered effects. Collectively, TLR2 mediates VEGF overproduction and downregulates ZO-1 expression in human PMCs, leading to mesothelial hyperpermeability in TBPE. Targeting TLR2/VEGF pathway may confer a potential treatment strategy for TBPE.

## 1. Introduction

Tuberculosis (TB) remains a crucial universal public health issue [1]. Tuberculous pleural infection, one of the most common extrapulmonary manifestations of TB, frequently complicates with pleural fluid exudation and fibrosis, resulting in pulmonary restriction and dyspnea [2].

Pleural mesothelium comprises a layer of pleural mesothelial cells (PMCs) that protect against invading microorganisms via Toll-like receptor (TLR) recognition of pathogen-associated molecular pattern (PAMP) and trigger downstream signaling with the production of numerous cytokines and chemokines, engaging in the pleural space inflammation [3]. TLR has been recognized as essential for the innate immune responses of PMCs in tuberculous pleural effusion (TBPE) [4,5]. Moreover, soluble TLR (sTLR), the extracellular domain of TLR, is shed upon infection and acts as a decoy receptor to inhibit PAMP-TLR engagement and downregulate the triggered inflammation [6]. Nevertheless, the process underlying the TLR-mediated inflammatory pleural fluid formation in TBPE remains to be explored.

PMCs adhere together by the intercellular junctional complexes, including adherens junctions, tight junctions, and desmosome, which fundamentally regulate the pleural mesothelial permeability [7]. Vascular endothelial growth factor (VEGF) is overproduced by stimulated PMCs [8,9,10], markedly elevated in exudative pleural effusions [11], and serves as a key factor in pleural fluid formation. VEGF increases vascular endothelial permeability by phosphorylating adherens junction proteins and disturbing the organization of tight junction proteins [12,13]. Moreover, VEGF downregulates the expression of the tight junction protein, zonula occludens-1 (ZO-1), in peritoneal mesothelial cells [14]. All the findings suggest that VEGF may alter pleural mesothelial permeability by disrupting the structural integrity of PMCs. However, the mechanisms implicit in pleural mesothelial hyperpermeability upon TB infection have rarely been investigated. In the present study, we aimed to evaluate the regulatory role of TLR2 in TB-triggered expression of VEGF and ZO-1 in PMCs and alteration in mesothelial permeability, as well as the clinical implication of soluble TLR2 (sTLR2) in TBPE.

## 2. Results

### 2.1. Levels of VEGF, Soluble TLR2, and Soluble TLR4 between Transudative and Tuberculous Pleural Effusions

A total of 52 patients diagnosed with transudative pleural effusion (TPE, n = 16) or TBPE (n = 36) were recruited (Table 1), inclusive of 30 male and 22 female individuals, ranging from 34 to 95 years of age. All TPE patients had a clinical diagnosis of heart failure. The pleural effusion size score, measured as the percentage of effusion shadowing in the hemithorax on chest radiograph (CXR), and the effusion values of pH and glucose were comparable between TPE and TBPE groups. In contrast, compared with TPE, the levels of lactate dehydrogenase (LDH) and adenosine deaminase, and the leukocyte count, were significantly higher in TBP. To identify the pathogenic role of VEGF and TLR signaling in TBPE, we measured VEGF, sTLR2, and soluble TLR4 (sTLR4) in pleural fluids, and found that effusion VEGF levels, sTLR2 and sTLR4 were substantially higher in TBPE than in TPE (Figure 1A–C). Furthermore, the effusion values of VEGF were positively correlated, while sTLR2 were negatively correlated, with pleural effusion size CXR score, in TBPE (Figure 1D,E). However, there was no significant relationship between sTLR4 and pleural effusion amount (Figure 1F). All these findings suggest that sTLR2 as well as VEGF are upregulated upon tuberculous pleural infection and that TLR2 signaling and VEGF are vitally involved in the development of TBPE.

### 2.2. Heat-Killed Mycobacterium Tuberculosis H37Ra Increased VEGF Expression in Human PMCs through JNK/AP-1 Signaling

Given that VEGF level was substantially elevated and positively correlated with pleural fluid volume in TBPE (Figure 1A,D), we next examined the mechanism of VEGF production in TBPE, employing the heat-killed *Mycobacterium tuberculosis H37Ra* (MTBRa) as a stimulant. We treated MeT-5A human PMCs with various concentrations of MTBRa for 24 h and evaluated VEGF protein expression by Western blot. The results showed that VEGF expression was significantly increased in PMCs upon exposure to 1 ng/mL MTBRa, compared with the resting condition (Figure 2A, Appendix A). Next, pharmacological signaling inhibitors of NF-κB (Parthenolide), PI3K/Akt (LY294002), MEK/ERK (PD98059), p38 (SB203580), and JNK (SP600125) were employed to examine which signaling pathway participated in MTBRa-induced VEGF expression in human PMCs. Cells were pretreated with each signaling inhibitor for 15 min, followed by exposure to MTBRa for 24 h. MTBRa-induced VEGF expression was significantly inhibited by 10 µM of JNK inhibitor (Figure 2B, Appendix A), while other signaling pathway inhibitors (20 µM) had no eminent effect on VEGF production. Consistently, MTBRa substantially activate JNK phosphorylation in PMCs as early as at 5 min, compared with the resting condition (Figure 2C, Appendix A). Moreover, MTBRa notably induced VEGF gene expression in PMCs in a dose- and time-dependent manner (Figure 2D), and JNK inhibitor considerably reduced MTBRa-induced VEGF mRNA expression (Figure 2E, Appendix A).

As JNK is the main upstream mediator that modulates activator protein-1 (AP-1) and downstream gene expression, we further examined the effect of MTBRa on the activation of the AP-1 constituent proteins (activating transcription factor 2 (ATF2) and c-Jun) in PMCs. We fractionated the cytoplasm and the nuclei of the cell lysates and then analyzed the amounts of phosphorylated c-Jun and ATF-2 in each fraction, using α-tubulin and YY1 as the marker proteins for the cytoplasmic and nuclear fractions, respectively. The cytoplasmic contamination in the nuclear fraction was determined by Western blotting (Appendix A). Compared with the resting condition, MTBRa remarkably increased cytoplasmic and nuclear c-Jun and ATF-2 phosphorylation within 30 min (Figure 3A,B), which were remarkably blocked by pretreatment with JNK inhibitor (Figure 3C–F). Collectively, these findings suggested that *Mycobacterium tuberculosis* (MTB) induces VEGF expression in PMCs via activation of JNK/AP-1 signaling.

### 2.3. MTBRa Stimulats Binding of AP-1 to VEGF Promotor in Human PMCs

Furthermore, we used the chromatin immunoprecipitation (ChIP)-quantitative PCR (qPCR) method to explore whether MTBRa stimulated the binding of AP-1 to VEGF promoter. The ChIP-qPCR analysis revealed that, as compared with IgG control, the c-Jun-binding DNA fragment was significantly enriched in MeT-5A cells upon MTBRa stimulation (Figure 4), indicating that MTBRa induces AP-1 DNA binding and initiates VEGF expression.

### 2.4. MTBRa and VEGF Downregulate ZO-1 Expression in Human PMCs

Subsequently, we investigated the modulation of MTB and the induced VEGF in pleural permeability by examining the effect of MTBRa and VEGF on ZO-1 expression in human PMCs. The results revealed that, compared with the resting group, MTBRa as well as VEGF significantly downregulated ZO-1 expression in a time-dependent manner, with the greatest effect occurring at 48 h (Figure 5A,B). Consistently, both MTBRa and VEGF concentration-dependently reduced ZO-1 expression in human PMCs, with the maximal effect occurring at 100 ng/mL MTBRa and 50 ng/mL VEGF, respectively (Figure 5C,D). Given that MTBRa significantly induced VEGF expression in PMCs (Figure 2), the present data may suggest that MTB inhibits ZO-1 expression to alter pleural permeability through the action of VEGF.

### 2.5. TLR2 Mediates Upregulation of VEGF and Downregulation of ZO-1 in MTBRa-Stimulated Human PMCs

Furthermore, to evaluate the role of TLR in VEGF overproduction and mesothelial hyperpermeability in TBPE, we examined the regulation of TLR on VEGF and ZO-1 expression in human PMCs upon MTBRa stimulation with the use of the TLR knockdown method. The TLR2 knockdown efficiency is shown in Appendix A. As shown in Figure 6A,B, transfection with scrambled or TLR2 siRNA did not influence the expression of phosphorylated JNK or VEGF in unstimulated PMCs. In contrast, the silencing of TLR2 significantly attenuated MTBRa-induced JNK phosphorylation and VEGF expression, as compared with scrambled controls. By contrast, knockdown of TLR4 did not substantially affect the increased JNK phosphorylation (Figure 6C) and VEGF expression (Figure 6D) in MTBRa-stimulated PMCs. These results indicated that TLR2, rather than TLR4, mediates upstream regulation of JNK activation and VEGF overproduction in TB-infected PMCs.

Furthermore, TLR2 silencing (Figure 7A) as well as pretreatment with a specific anti-VEGF neutralizing antibody (Figure 7B) considerably reversed the inhibition of ZO-1 expression by MTBRa. The immunofluorescence study (Figure 7C) further validated the upstream regulation of TLR2 and VEGF on ZO-1 in MTBRa-stimulated HPMCs. Collectively, the data suggested that TLR2 essentially mediates VEGF upregulation and subsequent ZO-1 downregulation in TB-infected PMCs.

### 2.6. MTBRa Induces Pleural Mesothelial Hyperpermeability through TLR2/VEGF Pathway

Given that TLR2 mediates VEGF upregulation and ZO-1 downregulation, we further explored the role of TLR2 in pleural mesothelial hyperpermeability in TBPE by using a transwell permeability assay. The results demonstrated that, comparable to VEGF (positive control), MTBRa significantly increased mesothelial permeability in a time-dependent manner, with the greatest effect occurring at 48 h (Figure 8A). Moreover, silencing of TLR2 and pretreatment with anti-VEGF antibody remarkably blocked the MTBRa-induced pleural hyperpermeability (Figure 8B), suggesting that MTBRa induces pleural mesothelial hyperpermeability through TLR2/VEGF pathway and that TLR2 signaling may play a key role in pleural fluid formation in TBPE.

Collectively, all these results indicate that TLR2 mediates MTB-activated cellular processes in human PMCs, by triggering JNK/AP-1 signaling to induce VEGF production and downregulate ZO-1 expression, leading to pleural mesothelial hyperpermeability and pleural fluid formation in TBPE (Figure 9). Moreover, effusion sTLR2 is significantly elevated and negatively correlated with pleural effusion size, suggesting that it may serve as a decoy receptor and reduce TLR2-mediated inflammation in TBPE.

## 3. Discussion

The current study revealed that the effusion levels of VEGF and sTLR2 were significantly elevated in TBPE compared to TPE. Moreover, effusion VEGF levels correlated positively, while sTLR2 values correlated negatively, with pleural effusion size in TBPE. In human PMCs, MTBRa substantially activated JNK/AP-1 signaling and upregulated VEGF expression, whereas knockdown of TLR2 remarkably inhibited MTBRa-induced JNK phosphorylation and VEGF overexpression. Additionally, both MTBRa and VEGF markedly reduced ZO-1 expression and induced pleural mesothelial permeability, while pretreatment with TLR2 siRNA or anti-VEGF antibody significantly attenuated the MTBRa-triggered effects. To our knowledge, this study is the first to demonstrate that TLR2 mediates JNK/AP-1 activation, VEGF overproduction, and ZO-1 downregulation in MTB-activated PMCs and to signify the clinical significance of TLR2/VEGF pathway in mesothelial hyperpermeability in TBPE.

The pathogenesis of pleural fluid formation in TBPE remains to be explored. Consistent with our previous report [15], the current data revealed that effusion VEGF was substantially elevated and positively correlated with pleural effusion size in TBPE, implying that VEGF plays a key role in MTB-triggered inflammation and pleural fluid exudation. VEGF has long been identified as a potent inducer of vascular permeability by inducing phosphorylation of cadherin and downregulation of occludin in vascular endothelial cells [12,13]. Nevertheless, the modulation effects of VEGF on mesothelial permeability and the intercellular junction between adherent PMCs are rarely investigated. Our in vitro experiments demonstrated that exogenous VEGF as well as MTBRa time- and concentration-dependently repressed ZO-1 expression in PMCs and induced pleural mesothelial permeability, and that anti-VEGF neutralizing antibody markedly abolished the permeability-inducing effects of MTBRa. These results are in line with the preceding observation in peritoneal mesothelial cells [14], and highly suggest that MTB enhance pleural mesothelial permeability through the action of VEGF.

Since the present study revealed the abundance of VEGF in TBPE, we conducted a mechanistic investigation of VEGF expression in MTB-infected PMCs. A previous report revealed that *Mycobacterium bovis* BCG stimulated the release and gene expression of VEGF in human PMCs [16]. Moreover, our previous work has demonstrated that MTB activated TLR2/ERK signaling to induce tumor necrosis factor (TNF)-α production in PMCs [4]. In the current study, we used the attenuated strain MTBRa as a stimulant of human PMCs and further explored the signal pathway underlying the VEGF synthesis. The data showed that MTBRa activated JNK/AP-1 signaling to elicit VEGF expression. In addition, silencing of TLR2, rather than TLR4, remarkably attenuated MTBRa-induced JNK phosphorylation and VEGF expression, highly indicating that MTBRa upregulates VEGF expression in human PMCs via TLR2/JNK/AP-1 pathway. To our knowledge, this study is the first to delineate the involvement of TLR2-mediated signaling in VEGF overproduction by human PMCs upon MTB infection.

Accordingly, our research again validated the key regulation of TLR2 on the immune response against MTB infection in PMCs [4,5]. At the same time, growing evidence showed that sTLR2 functions as a decoy receptor and significantly attenuated the TLR2-mediated immune inflammatory responses [6,17,18]. To verify the clinical implication of soluble TLR, we measured the pleural fluid levels of sTLR2 and sTLR4 in TBPE patients. In parallel with previous reports [19,20], our results demonstrated significantly higher levels of sTLR2 and sTLR4 in TBPE than in TPE, implying the activation of TLR signaling and substantial release of sTLR2 and sTLR4 in the pleural space upon MTB infection. Moreover, the effusion levels of sTLR2, but not TLR4, correlated negatively with the effusion size of TBPE, which may indicate that sTLR2 mitigates MTB-triggered, TLR2-mediated mesothelial hyperpermeability and pleural fluid formation. Thus, all the current results feature that TLR2 is essential for the production of inflammatory pleural effusion and sTLR2 may serve as a marker for TLR-signaling activation and clinical severity in TBPE.

As previous and present studies have indicated that TLR2 and VEGF are essentially implicated in MTB-induced mesothelial hyperpermeability and pleural fluid exudation [12,13,14,15,16], employing TLR2 or VEGF inhibition as a therapeutic strategy for TBPE is merited. Inhibition of TLR2 signaling with antibodies [21], small molecules [22], synthetic decoy peptides [23], or soluble TLR2 [24] has been shown to decrease damaging inflammation and improve disease outcome in preclinical studies [25]. Moreover, anti-VEGF antibody significantly reduced inflammatory effusion in vivo and malignant or nonmalignant effusion in humans [26,27,28]. Therefore, further studies targeting TLR2 and VEGF expression in pleural mesothelium are warranted to develop innovative adjunctive treatments for TBPE.

However, the current study has some limitations. First, although sTLR2 levels correlate significantly with the effusion size of TBPE, its real impact on clinical outcomes remains unknown. Further large-scale human studies are commanded to validate whether effusion sTLR2 can be used as a predictive, prognostic, or therapeutic biomarker for TBPE. Second, although TLR2 essentially mediates inflammatory pathways and mesothelial hyperpermeability, the treatment potential of targeting TLR2 on TBPE is yet to be explored. Further in vivo experiments are required to investigate the therapeutic effects of modulation of TLR2 signaling on TBPE. Nevertheless, to our knowledge, this is the first study featuring the cellular as well as the clinical implication of the TLR2 pathway in TBPE and may motivate future research on TLR2-targeted therapeutics for infectious pleural diseases.

## 4. Materials and Methods

### 4.1. Materials

Heat-killed *Mycobacterium tuberculosis* H37Ra (MTBRa) was purchased from Sigma. Recombinant VEGF were acquired from Peprotech (Cranbury, NJ, USA). Human VEGF, sTLR2, and sTLR4 Quantikine enzyme-linked immunosorbent assay (ELISA) kits were attained from R&D Systems (Minneapolis, MN, USA). Scrambled control, smart-pool TLR siRNAs, or transfection reagent were bought from Dharmacon (Lafayette, CO, USA). All antibodies were procured from GeneTex (Irvine, CA, USA) except that SB203580, SP600125, PD98059, LY294002, and parthenolide were acquired from Calbiochem (St. Louis, MO, USA).

### 4.2. Patient Enrolment

Patients presenting with pleural effusions were eligible and recruited if the diagnosis of TBPE or transudative pleural effusion (TPE) was established. TBPE was diagnosed by the detection of granulomatous inflammation with or without acid-fast bacilli on closed pleural biopsy specimens. Ethical approval was conferred by the Institutional Review Board of Taipei Medical University Hospital, Taipei, Taiwan (No. N201801026). Patients having bleeding tendency, receiving anticoagulant therapy, or with invasive pleural procedure history were excluded.

### 4.3. Thoracentesis and Pleural Fluid Analysis

Immediately, or within 24 h after admission, 50 mL of pleural fluid was collected by thoracentesis following the granting of informed consent. Pleural fluid analyses, adenosine deaminase, and microbiological studies were conducted as routine.

### 4.4. Pleural Effusion Size Chest Radiograph Score

Posterior–anterior chest radiography (CXR) was taken on admission and saved as a digital file in the picture archiving and communication system (PACS). The area of the pleural effusion opacity and the hemithorax were measured using the PACS image-processing program. Two independent radiologists who were blinded to all clinical data read each CXR to determine the estimated portion of pleural opacity in the hemithorax as the pleural effusion size CXR score [4].

### 4.5. Measurement of VEGF and Soluble TLRs in Pleural Fluids

The levels of VEGF, sTLR2, and sTLR4 in the supernatants of the acquired pleural fluids were measured by the commercially available ELISA kits as previously described [4].

### 4.6. Human Pleural Mesothelial Cell (PMC)

MeT-5A cells (#CRL-9444TM, ATCC, Manassas, VA, USA), a human pleural mesothelial cell line, were grown in medium 199 (GIBCO, Invitrogen, San Diego, CA, USA) as previously described [4]. The cells were subcultured every 4 days by TrypLE Express reagents. On starting the experiment, MeT-5A cells were seeded into 6 cm dishes. Upon reaching confluence, cells were cultured in serum-free medium for one day, then subjected to the indicated treatments.

### 4.7. Western Blot

The proteins in total cell lysates (30 µg) were separated by SDS-PAGE gel and transferred to nitrocellulose membranes. The blotting membranes were developed with individual primary antibodies, followed by horseradish peroxidase-conjugated secondary antibodies. The quantitative densitometric analysis was conducted by ChemiDoc MP Imaging System (BioRad, Hercules, CA, USA).

### 4.8. Reverse Transcription-Polymerase Chain Reaction (RT-PCR)

Total cellular RNA was extracted from MeT-5A cells using the TRIsure^®^ reagent (Bioline, London, UK) and 1 µg of RNA was used for cDNA synthesis (Super Script On-Step RT-PCR system, Thermo Fisher Scientific, Waltham, MA, USA). Amplification of human VEGF and GAPDH was performed using the following specific primers (sense/antisense): 5′-AGGGCAGAATCATCACGAAG-3′/5′-TTTAACTCAAGCTGCCTCGC-3′ and 5′-GCCGCCTGGTCACCAGGGCTG-3′/5′-ATGGACTGTGGTCATGAGCCC-3′, respectively. The PCR was performed as previously described [4].

### 4.9. Nuclear and Cytoplasmic Proteins Extraction

Minute ™ Cytosolic and Nuclear Extraction Kit (Invent Biotechnologies, Eden Prairie, MN, USA) was used to isolate nuclear and cytosolic proteins from cultured cells. Protein separation was confirmed by Western blot and YY1 and α-tubulin were used as a marker for nuclearand cytoplasmic proteins, respectively.

### 4.10. Chromatin Immunoprecipitation-Quantitative PCR (ChIP-qPCR)

The ChIP assay was performed using a Pierce Magnetic ChIP Kit (Thermo Scientific, Waltham, MA, USA) according to the manufacturer’s protocol. Briefly, 5 × 10^6^ MeT-5A cells were collected and crosslinked with 1% formaldehyde and then disrupted by ultrasonication. Specific antibodies to the protein of c-Jun (Abcam, Cambridge, MA, USA) and normal rabbit IgG, a negative control, were added to bind target protein-DNA complexes. Protein A agarose was added to bind the antibody-target protein-DNA complexes, and then the enriched target protein-DNA complexes were eluted from the beads and the crosslinks were reversed. After purification, enriched DNA fragments were subjected to qPCR analysis using the following primers: VEGF, AAGAACTCGGACCTCCTCAC (sense), and CGTTGCTGGACTGGATTATCA (antisense). qPCR analysis was performed using fast SYBR Green master mix (Applied Biosystems) following the manufacturer’s instructions. The following cycling conditions were used: 50 °C for 2 min, 95 °C for 10 min, 40 cycles of 95 °C for 15 s and 60 °C for 1 min, and then 95 °C for 1 sec. The fluorescence was measured using the QuantStudioTM 5 real-time PCR system (Applied Biosystems, Waltham, MA, USA). The fold change in the amount of the DNA fragment enriched by a specific antibody versus the total input was calculated by the 2^−ΔΔ^ Ct method [29].

### 4.11. RNA Interference

MeT-5A cells were transfected with a siRNA against TLR2 or TLR4, or scrambled siRNA for 24 h, using the DharmaFECT^®^ siRNA transfection reagent (Thermo Scientific, Waltham, MA, USA). The media were replaced with fresh serum-free M199 for another 24 h. Then cells were stimulated with MTBRa in various experiments and the effects on the expression of indicated proteins were assayed by Western blot.

### 4.12. Mesothelial Permeability

We used a transwell permeability assay for measurement of pleural permeability, with 5 × 10^5^ PMCs in 300 µL of M199 medium seeded on top of Transwell inserts in 24-well plates (3.0-µm pore size, Corning, Lowell, MA, USA) and grown to confluence. Then, 2 µL of streptavidin-horseradish peroxidase (HRP) was added to the upper chamber in the presence of fresh serum-free medium, MTBRa, VEGF, MTBRa plus scrambled siRNA or TLR2 siRNA, or MTBRa plus anti-VEGF antibody, and the samples incubated at 37 °C for the indicated times. After reaction, 100 µL aliquots from the lower chamber were transferred to a 96-well plate and then 50 µL 3,3′,5,5′-tetramethylbenzidine (TMB) substrate were added to measure HRP activity. Sampling reactions were stopped by the addition of 25 µL of stop solution and the absorbance at 450 nm was measured using a spectrophotometer.

### 4.13. Immunofluorescence Staining

MeT-5A cells were seeded on a collagen-coating glass. After treatment, cells were fixed with 4% paraformaldehyde and permeabilized with 0.2% Triton X-100 for 20 min. Afterward, cells were blocked for 30 min in 3% bovine serum albumin at room temperature and incubated overnight at 4 °C with specific primary antibodies. Cells were then incubated with FITC-conjugated goat anti-rabbit secondary antibody for 1 h and subsequently counterstained with 4′,6-diamidino-2-phenylindole (DAPI). Stained cells were mounted with FluoroGel mounting medium (Interchim, San Diego, CA, USA) and visualized under a confocal laser scanning microscope.

### 4.14. Statistical Analyses

Results were presented as mean ± SEM or median (range). One-way analysis of variance was used to determine if significant differences existed among the three groups. Data comparisons between any two groups were conducted employing an unpaired t-test or the Mann–Whitney U method where appropriate. A two-tailed *p*-value < 0.05 was considered statistically significant.

## 5. Conclusions

In conclusion, TLR2 mediates VEGF overproduction and downregulates ZO-1 expression in human PMCs, leading to mesothelial hyperpermeability and pleural fluid formation in TBPE. Targeting TLR2/VEGF pathway may confer a potential treatment strategy for TBPE.

## Figures and Tables

**Figure 1 ijms-24-02846-f001:**
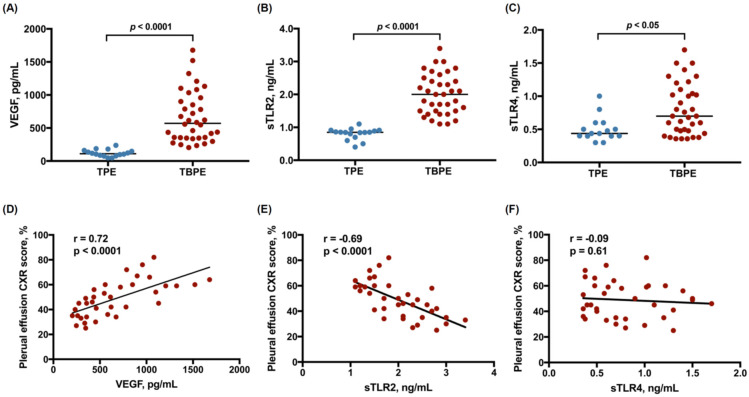
Levels of VEGF, sTLR2, and sTLR4 between TPE and TBPE (**A**–**C**) and their correlation with pleural effusion CXR score in TBPE patients (**D**–**F**). TPE, transudative pleural effusion (n = 16); TBPE, tuberculous pleural effusion (n = 36); VEGF, vascular endothelial growth factor; sTLR2, soluble Toll-like receptor 2; sTLR4, soluble Toll-like receptor 4.

**Figure 2 ijms-24-02846-f002:**
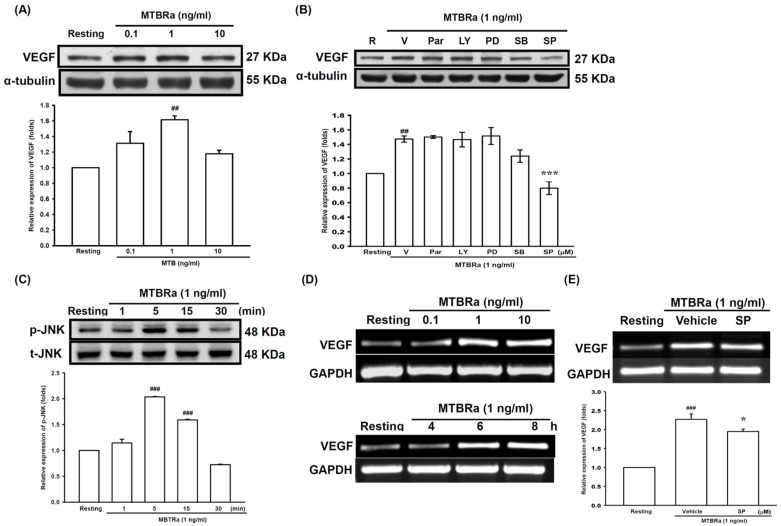
MTBRa induces VEGF expression in human PMCs through JNK signaling. (**A**) MeT-5A cells were treated with MTBRa (0.1–10 ng/ml) for 24 h. VEGF protein was assayed by Western blotting. ^##^ *p* < 0.01 compared with resting group; (**B**) MeT-5A cells were pretreated with vehicle (V), Parthenolide (Par, 20 µM), LY294002 (LY, 20 µM), PD98059 (PD, 20 µM), SB203580 (SB, 20 µM), or SP600125 (SP, 10 µM), then treated with MTBRa (1 ng/ml) for 24 h. VEGF protein was examined by Western blotting. **^##^** *p* < 0.01 compared with resting group, *** *p* < 0.001 compared with vehicle group; (**C**) MeT-5A cells were treated with MTBRa (1 ng/ml) for the specified times, and JNK phosphorylation was assessed by Western blotting. ^###^ *p* < 0.001, compared with resting group; (**D**) MeT-5A cells were treated with various concentrations of MTBRa (0.1–10 ng/ml) for 8 h or with 1 ng/ml MTBRa for the indicated times. The levels of VEGF mRNA were measured by RT-PCR. (**E**) MeT-5A cells were pretreated with vehicle or SP 10 µM for 15 min, followed by MTBRa treatment. The VEGF mRNA expression was quantified by RT-PCR. ^###^ *p* < 0.001 compared with resting group; * *p* < 0.05 compared with vehicle group. The relative folds of densitometrical data of all studies were expressed as mean ± SEM of three independent experiments. VEGF, vascular endothelial growth factor; PMC, pleural mesothelial cell; MTBRa, *Mycobacterium tuberculosis* H37Ra; JNK, c-Jun N-terminal kinases; GAPDH, glyceraldehyde 3-phosphate dehydrogenase.

**Figure 3 ijms-24-02846-f003:**
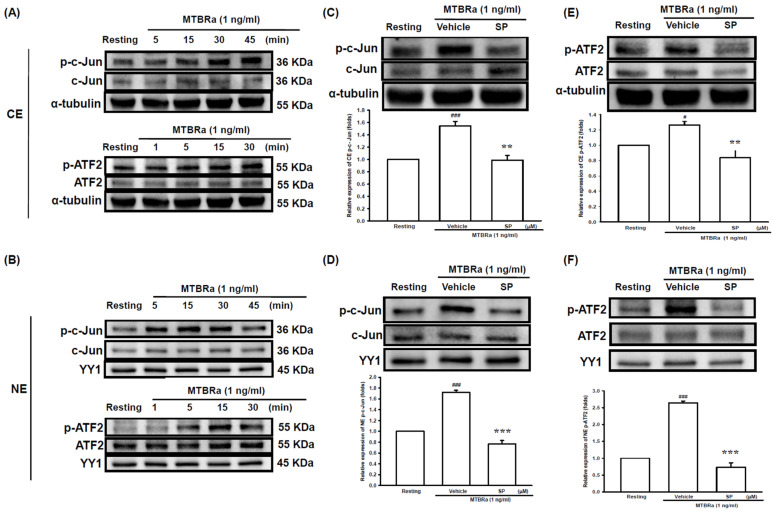
MTBRa activates JNK/AP-1 signaling in human PMCs. Western blotting was used to measure phosphorylated c-Jun and ATF-2 in (**A**) cytoplasmic and (**B**) nuclear extracts in MeT-5A cells treated with 1 ng/ml MTBRa for the indicated times. Western blotting was used to measure the amount of phosphorylated c-JUN in (**C**) cytoplasmic or (**D**) nuclear extract in MeT-5A cells pretreated with vehicle or SP 10 µM followed by MTBRa treatment for 15 min. Western blotting was used to measure the amount of phosphorylated c-ATF2 in (**E**) cytoplasmic or (**F**) nuclear extract in MeT-5A cells pretreated with vehicle or SP 10 µM followed by MTBRa treatment for 15 min. The relative folds of densitometrical data of all studies were expressed as mean ± SEM of three independent experiments. **^#^** *p* < 0.05, **^###^** *p* < 0.001 compared with resting group; ** *p* < 0.01, *** *p* < 0.001 compared with vehicle group. MTBRa, *Mycobacterium tuberculosis* H37Ra; PMC, pleural mesothelial cell; SP, SP600125; CE, cytoplasmic extract; NE, nuclear extract.

**Figure 4 ijms-24-02846-f004:**
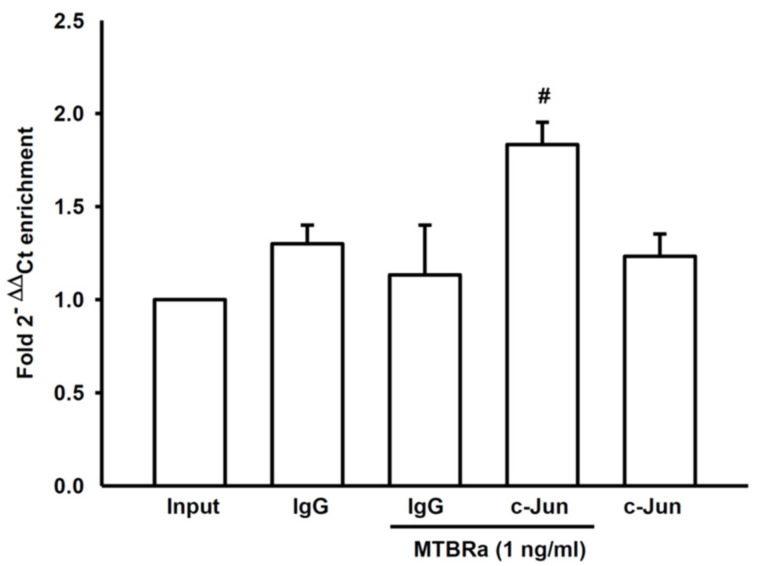
MTBRa stimulates the binding of AP-1 to the promoter of VEGF in human PMCs. ChIP-qPCR analysis of the binding of AP-1 (c-Jun) to the genomic region of VEGF with input normalization and IgG control in MeT-5A cells treated with MTBRa for 15 min. The relative fold enrichment data were expressed as mean ± SEM of three independent experiments. **^#^** *p* < 0.05 compared with MTBRa-treated IgG control group. MTBRa, *Mycobacterium tuberculosis* H37Ra; PMC, pleural mesothelial cell; VEGF, vascular endothelial growth factor; ChIP-qPCR, chromatin immunoprecipitation-quantitative PCR.

**Figure 5 ijms-24-02846-f005:**
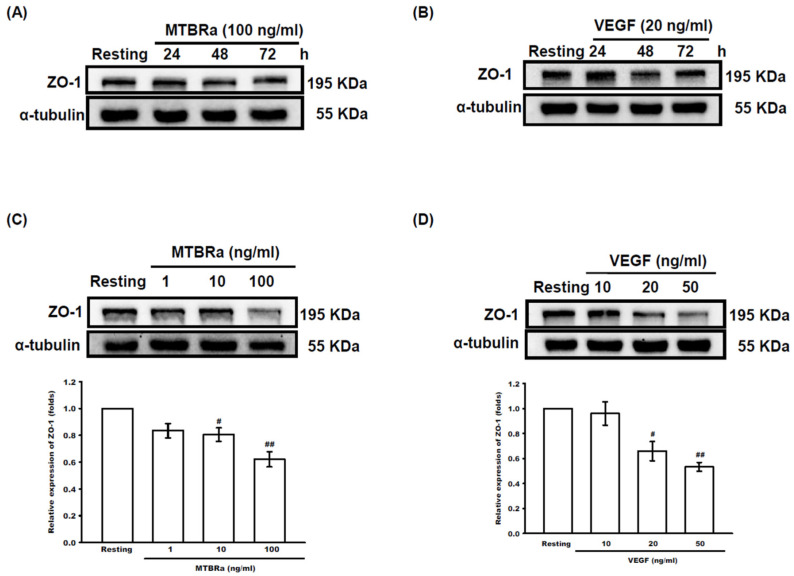
MTBRa and VEGF downregulate ZO-1 expression in human PMCs. MeT-5A cells were treated with (**A**) MTBRa and (**B**) VEGF for the indicated times or stimulated with various concentrations of (**C**) MTBRa (1–100 ng/mL) and (**D**) VEGF (10–50 ng/mL) for 48 h. ZO-1 expression was analyzed by Western blotting. Data represented three independent experiments. ^#^ *p* < 0.05, ^##^ *p* < 0.01 compared with resting group. MTBRa, *Mycobacterium tuberculosis* H37Ra; VEGF, vascular endothelial growth factor; ZO-1, zonula occludens; PMC, pleural mesothelial cell.

**Figure 6 ijms-24-02846-f006:**
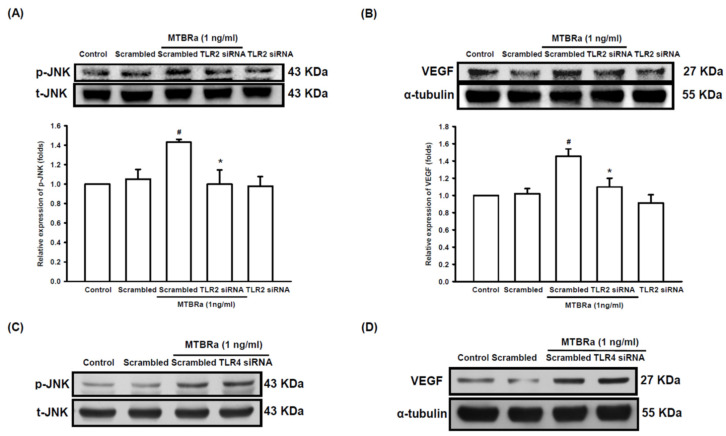
MTBRa activates TLR2/JNK pathway to increase VEGF expression in human PMC cells. MeT-5A cells were transfected with scrambled siRNA and (**A**,**B**) TLR2 siRNA (25 nM) or (**C**,**D**) TLR4 siRNA (25 nM) for 48 h, then treated with or without MTBRa stimulation. JNK phosphorylation and VEGF expression were determined by Western blotting after MTBRa stimulation for 15 min and 24 h, respectively. Data are representative of three separate experiments. ^#^ *p* < 0.05 compared with the scrambled siRNA group; * *p* < 0.05 compared with MeT-5A cells transfected with scrambled siRNA, followed by MTBRa treatment group. MTBRa, *Mycobacterium tuberculosis* H37Ra; VEGF, vascular endothelial growth factor; PMC, pleural mesothelial cell.

**Figure 7 ijms-24-02846-f007:**
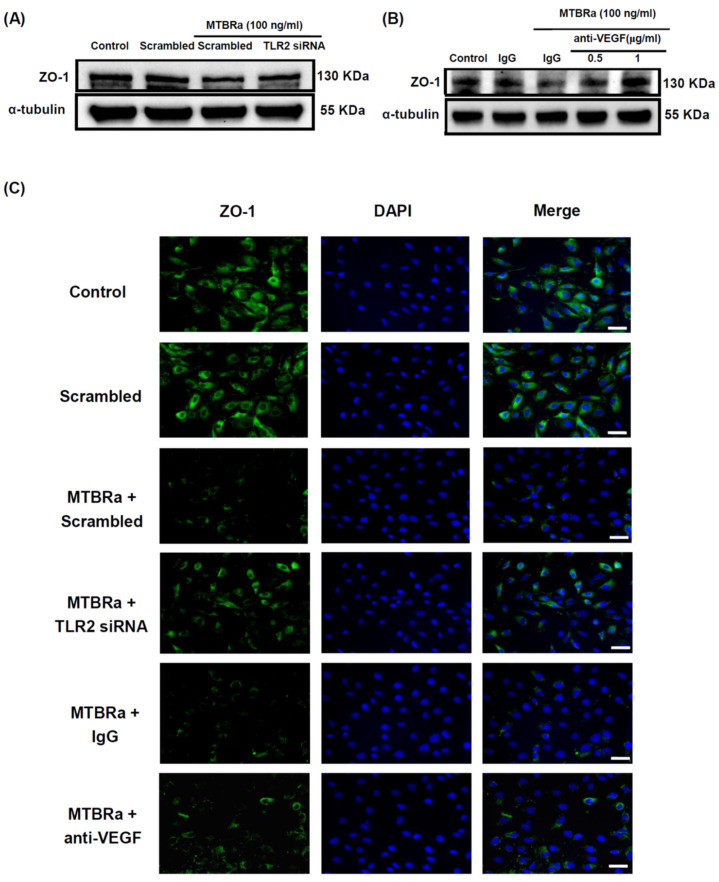
MTBRa downregulates ZO-1 expression through TLR2/VEGF signaling in human PMCs. MeT-5A cells were transfected with scrambled siRNA or TLR2 siRNA for 48 h or pretreated with IgG isotype control or anti-VEGF neutralizing antibody for 15 min, followed by MTBRa treatment for 24h. Expression of ZO-1 was assayed by (**A**,**B**) Western blotting or (**C**) confocal immunofluorescence analysis (green). Cells were co-stained with 4′,6-diamidino-2-phenylindole (DAPI) for exhibition of the nuclei (blue). Scale bar, 30 µm. Data represented three individual experiments. MTBRa, *Mycobacterium tuberculosis* H37Ra; PMC, pleural mesothelial cell.

**Figure 8 ijms-24-02846-f008:**
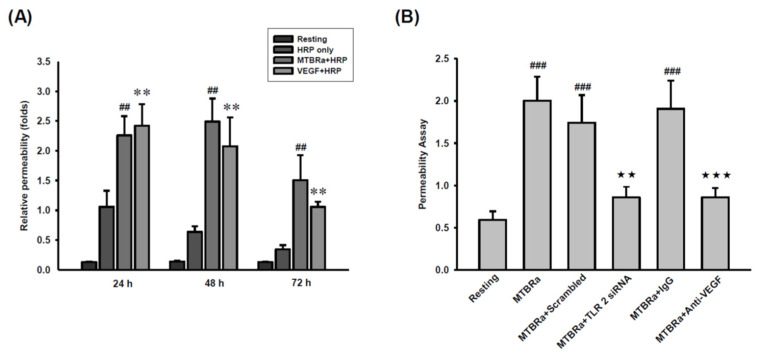
MTBRa induces pleural mesothelial hyperpermeability through TLR2/VEGF pathway. (**A**) MeT-5A cells were treated with HRP, MTBRa plus HRP, or VEGF plus HRP for the indicated times. (**B**) MeT-5A cells were transfected with scrambled or TLR2 siRNA, or pretreated with IgG isotype control or anti-VEGF neutralizing antibody, then stimulated with MTBRa for 48 h. The pleural mesothelial permeability was measured by a transwell permeability assay, as described in Section 4. Data represented three independent experiments. ^##^ *p* < 0.01, ^###^ *p* < 0.001 compared with the resting group; ** *p* < 0.01 compared with cells treated with HRP only group. ^★★^ *p* < 0.01 compared with cells transfected with scrambled siRNA, followed by MTBRa treatment group; ^★★★^ *p* < 0.001 compared with cells pretreated with IgG isotype and treated with MTBRa group. Data are representative of three independent experiments. HRP, streptavidin-horseradish peroxidase; MTBRa, *Mycobacterium tuberculosis* H37Ra.

**Figure 9 ijms-24-02846-f009:**
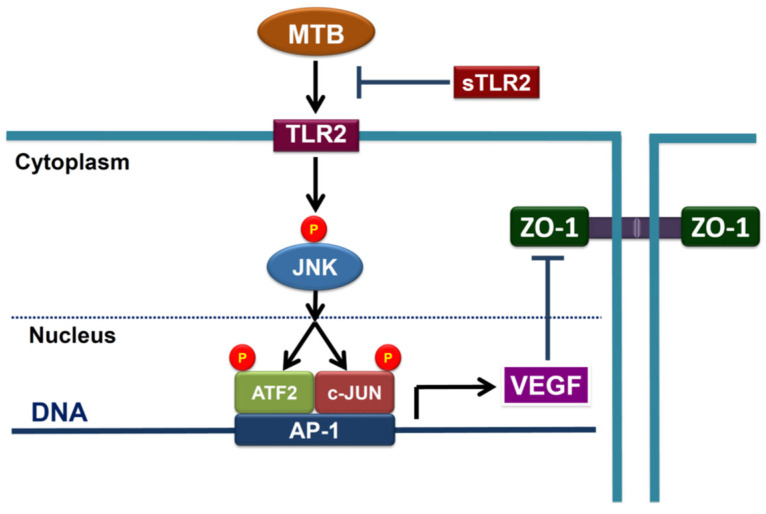
Schematic diagram of the cellular mechanism underlying pleural mesothelial hyperpermeability in tuberculous pleural effusion. TLR2 mediates *Mycobacterium tuberculosis*-activated cellular processes in human pleural mesothelial cells by triggering JNK/AP-1 signaling to induce VEGF production and thereby downregulate ZO-1 expression, leading to pleural mesothelial hyperpermeability. Moreover, sTLR2 serves as a decoy receptor to reduce TLR2-mediated inflammation. MTB, *Mycobacterium tuberculosis*; TLR2, Toll-like receptor 2; sTLR2, soluble Toll-like receptor 2; JNK, c-Jun N-terminal kinases; ATF2, activating transcription factor 2; AP-1, activator protein-1; VEGF, vascular endothelial growth factor; ZO-1, zonula occludens.

**Table 1 ijms-24-02846-t001:** Demographics and Pleural Fluid Characteristics *.

	TPE	TBPE	*p*-Value
Subjects, n	16	36	
Age, years	83 (54–95)	78 (34–90)	0.058
Male, n (%)	9 (57)	21 (58)	0.888
Pleural effusion CXR score, %	53 (46–65)	46 (19–87)	0.107
Pleural fluid			
pH value	7.38 (7.32–7.42)	7.36 (6.90–7.51)	0.084
Glucose, mg/dL	120 (73–152)	122 (16–188)	0.691
LDH, IU/dL	118 (57–135)	293 (64–1999)	<0.0001
Leukocyte count, cells/uL	354 (115–490)	1044 (81–15,840)	0.001
ADA, IU/L	25 (6–39)	124 (48–262)	<0.0001

Abbreviation: TPE, transudative pleural effusion; TBPE, tuberculous pleural effusion; CXR, chest radiograph; LDH, lactate dehydrogenase; ADA, adenosine deaminase. * Values are presented as median (range) unless otherwise specified.

## Data Availability

The data presented in this study are available on request from the corresponding author.

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
