# Peer review of "Toll-like Receptor 2 Mediates VEGF Overexpression and Mesothelial Hyperpermeability in Tuberculous Pleural Effusion"

_ijms, 2023, doi:10.3390/ijms24032846_

Round 1

Reviewer 1 Report

This manuscript by Wei-Lin Chen and coworkers reports the mechanism of pleural mesthelial hyperpermeability in TB pleural effusion.  They start with patient data and then use pleural mesothelial cells and the MeT-5A cell line to figure out the mechanisms involved using a heat killed preparation of MTB H37Ra.  Overall, the paper was easy to read and follow, the results and figures were explained well, and outlay of the paper was logical. However, there were few things I was skeptical about or which require some clarification which I want to point out:

·         Some of the conclusions based in Fig. 2 and 3 depend on accurate densitometric data.  Since the visual data is very close and seemingly insignificantly different, the authors should provide the raw densitometric data in a supplementary file for critical review by the reader.  There is an issue with pixel saturation.  It would be helpful to know if the data was collected on film and then measured by the ChemiDoc or was the HRP illumination directly measured by ChemiDoc.

·         In fig 3, the authors were using 1ng/ml MTBRa treatment for inducing p-c-Jun and p-ATF2. In fig 4,  why was the dose increased to 100ng/ml MTBRa treatment is done despite using the same cell types. In fig 4c 100ng/ml of MTBRa concentration is significantly reducing ZO-1 expression which is different from 1ng/ml of MTBRa working concentration used in Fig 2 and 3.

·         Why is 7a the only colored bar graph?

·         The second figure 7 needs to be labeled Fig. 8.

·         The discussion may want to point out some of the limitations with the current work, but otherwise, hits all the necessary points of why this report is significant.

Reviewer 2 Report

The authors of the manuscript "Toll-like Receptor 2 Mediates VEGF Overexpression and Mesothelial Hyperpermeability in Tuberculous Pleural Effusion" have performed interesting analysis for the role of TLR2 in regulating the VEGF expression during Tuberculous Pleural Effusion. The results are promising but the figure quality in the manuscript is very poor. Below are some of the experiments required in order to support some of the conclusions made by the authors and improve data quality.

1.    In figure 2a, b, and c please label above each lane the sample name instead of only the box plot below. Please add molecular weights for all the western blots in the manuscript. Increase the axis font size since in the current form they are not legible.

2.    In figure 3, the conclusions made by the authors about the nuclear translocation of c-Jun and ATF2 are not supported by only the current data. The authors needs to have one lane of whole cell/ cytoplasmic protein fraction in their western blot along with their nuclear protein fractions, in order to show the purity of their nuclear fractions. Also, probe the blot against a cytoplasmic protein to show that there is no cytoplasmic contamination in your nuclear fractions.

3.    In figure 3, are the total c-Jun and ATF2 levels increasing upon exposure to MTBRa? This data is missing and therefore the conclusion about increased phosphorylation of these proteins into p-c-Jun and p-ATF2 is not yet conclusive. Also, please add labels over each lane in 3b and 3c.

4.    In figure 5a and b, please add again the labels to each lane. Here one important control is missing which is the TLR2 siRNA alone without any MTBRa exposure, since the authors claim in line 192-193 that “TLR2 essentially mediates VEGF upregulation and subsequent ZO-1 downregulation in TB-infected PMCs”. The control will help us know whether TLR2 can downregulate p-JNK and VEGF even in the absence of MTBRa.

5.    In figure 5, the knockdown efficiency of TLR2 is not shown. This data needs to be added.

6.    In figure 7, in the schematic diagram the authors have suggested that p-ATF2 and p-c-Jun bind DNA to regulate VEGF expression. However, it would be really great for the impact of the manuscript if the authors could add ChIP-qCPR data for these proteins and show that indeed upon MTBRa exposure they bind VEGF promoter more. This will be an important data since in the current form the nuclear translocation of these proteins in not conclusive (comment 3 above).

Minor point:

1.    In the legends for 2b, please avoid calling it VEGF expression since you are measuring protein levels and not gene expression.

I believe that the manuscript has merit however some of their conclusions need more experimental support, therefore, I urge the authors to clarify these queries raised above.

Round 2

Reviewer 2 Report

The authors of the manuscript "Toll-like Receptor 2 Mediates VEGF Overexpression and Mesothelial Hyperpermeability in Tuberculous Pleural Effusion" have addressed most of my review comments.

The addition of the proper controls (siRNA efficiency, total protein levels of c-Jun and ATF2) make the results presented by authors more convincing. The ChIP-qPCR data also shows that the c-Jun binding to VEGF promoter is MTBRa stimulated, further supporting authors claims.

Therefore, I see that the manuscript has improved extensively and is fit for publication after minor spell checks.